# A Note on Connectivity and Stability in Dynamic Network Formation

**Tackseung Jun and Jeong-Yoo Kim ***

Department of Economics, Kyung Hee University, 1 Hoegidong, Dongdaenumku, Seoul 130-701, Korea; tj32k@khu.ac.kr
**\*** Correspondence: jyookim@khu.ac.kr; Tel./Fax: +82-2-961-0986

**Abstract:** We consider the dynamic network formation problem under the requirement that the whole network be connected and remain connected after $q$ nodes are destroyed. We propose the concept of dynamic $\mathscr{C}^q$-stability and characterize dynamic $\mathscr{C}^q$-stable networks for any $q \geq 0$. Comparison with the outcome in the static model is also discussed.

**Keywords:** connectivity; network formation; dynamic $\mathscr{C}^q$-stability

## 1. Introduction

The value of a social network relies crucially on its connectivity and resilience, which can be defined by the ability to maintain connectivity even after unexpected trembles due to external shocks. In a prior paper [1], we considered the problem of forming a network under the requirement that the whole network be connected and remain connected after eliminating $q$ nodes, and introduced the concept of the $\mathscr{C}^q$-stability of a network in a static setting. The $\mathscr{C}^q$-stability is, roughly speaking, to require connectivity in addition to pairwise stability defined by Jackson and Wolinsky [2] even after any at most $q$ nodes are deleted. In the paper, we characterized the $\mathscr{C}^1$-stable networks without characterizing general $\mathscr{C}^q$-stable networks.

In this paper, we extend the network formation problem and the concept of $\mathscr{C}^q$-stability to a dynamic setting. In a dynamic model in which one player is spawned at each period, we characterize the dynamic $\mathscr{C}^q$-stable network for $q \geq 0$ and compare the outcome with that in the static model of Jun and Kim [1]. Our concept of dynamic $\mathscr{C}^q$-stability will be also called intraperiod-farsighted stability, since it is defined based on our assumption that players are forward looking only within each period.

There are some papers on the resilience of a network. To the best of our knowledge, Jun and Kim [1] is the first paper to acknowledge the possibility of node destruction in the literature on (social or economic) network formation. After Jun and Kim [1], there have been some studies that added an attacker as an active player. In particular, Goyal and Vigier [3] considered a contest game between a network designer and an attacker. The designer first chooses a network and allocates defense resources across nodes, and then an adversary allocates attack resources to nodes. If the adversary's attack on a node succeeds, he captures the node and any unprotected nodes connected with the nodes via contagion. Then, the designer cannot control the nodes anymore. The objective of the designer is to maximize the total value from the whole network. In Dziubiński and Goyal [4], there is no contagion but the model is quite similar. Unlike Jun and Kim [1], however, both are centralized network design problems rather than decentralized network formation problems. Moreover, information decay is assumed in neither model. Thus, in Goyal and Vigier [3], a network generates the same value insofar as it is connected. That is, how it is connected does not matter. In Dziubiński and Goyal [4] that considers



the linking costs explicitly, it is not important, either, how a network is connected and, as a result, how much it can reduce communication delays, because information decay is not assumed. Hoyer and De Jaegher [5] also considers a centralized network design problem with linking costs and without information decay. They analyze both cases of node deletion and link deletion. Exceptionally, Haller and Hoyer [6] is a decentralized network formation model just like Jun and Kim [1], but it assumes no information decay, either. It uses the solution concept of Nash stability, whereas Jun and Kim [1] proposes a new solution concept of $\mathscr{C}^q$-stability which shares the spirit of pairwise stability but is stronger than that. It is well known that Nash stability and pairwise stability do not have any inclusion relation. On one hand, Nash stability could be stronger than pairwise stability because it requires a network to be robust against unilateral deviations involving more than two players (for example, cutting a link and at the same time linking with the third player) and only a pair's deviation, but, on the other hand, pairwise stability could be stronger because it requires a network to be robust against any bilateral deviations of a pair of players to form a link and a unilateral deviation.

Our dynamic model of network formation is also similar to a sequential version of a network creation game which was introduced by Fabrikant et al. [7]. Both take the delay cost into account but Fabrikant et al. [7] used Nash stability as a solution concept, unlike our paper. Leme et al. [8] proposed subgame perfection as the solution concept in a model in which players arrive sequentially, just as we used the concept of intraperiod-farsighted stability to capture dynamic consistency, but neither of Fabrikant et al. [7] and Leme et al. [8] considered the possibility of attacks.

In Section 2, we introduce some preliminary definitions in graph theory. In Section 3, we set up a dynamic model of network formation. Section 4 provides an analysis for the dynamic model without faults and with faults. Concluding remarks follow in Section 5. All the proofs are provided in Appendix A.

## 2. Preliminaries

Let $N = \{1, 2, \cdots, n\}$ be the set of nodes representing agents with $n \geq 1$. Interactions among agents are formally represented by a graph (or network) $g$.[1]

If agents $i$ and $j$ are linked to each other in graph $g$, we denote it by $ij \in g$. We also denote by $g + ij$ the graph obtained from adding the link $ij$ to $g$ and by $g - ij$ the graph obtained from deleting the link $ij$ from $g$. A path between $i_1$ and $i_n$ in $g$ is a set of distinct nodes $\{i_1, i_2, \cdots, i_n\} \subset N$ such that $i_1 i_2, i_2 i_3, \cdots, i_{n-1} i_n \in g$. We will use more intuitive notation for a path as $i_1 \rightarrow i_2 \rightarrow \cdots \rightarrow i_{n-1} \rightarrow i_n$ interchangeably.

A graph $g$ is said to be connected if there is a path in $g$ between any pair of agents $i$ and $j$ in $N$. Additionally, $g$ is said to be $q$-connected if $g$ is still connected even after $q$ nodes are deleted.[2] We will denote the set of all connected graphs by $\mathbb{C}$ and the set of all $q$-connected graphs by $\mathbb{C}^q$. Then, it is obvious that graph $g$ is $q$-connected whenever it is $q'$-connected for $q < q'$, that is, $\mathbb{C}^{q'} \subset \mathbb{C}^q$ if $q < q'$ for all $q = 0, 1, \cdots, n$ where $\mathbb{C}^0 \equiv \mathbb{C}$.

If $g$ is not connected, it is possible that only a proper subset of $N$ have links in $g$. We will denote the subset by $S(g)$ and define it by $S(g) = \{i, j \in N \mid ij \in g\}$.

## 3. Dynamic Model of Network Formation

We consider communication networks. If $ij \in g$, both of agent $i$ and agent $j$ incur the cost $c \in (0, \infty)$ of connecting the direct link. If agent $i$ and $j$ are directly linked, they can exchange information without delay (i.e., in one unit of time length). If agent $i$ is indirectly linked to $j$, i.e., there is a path $\{i, i_1, \cdots, i_k, j\}$ between $i$ and $j$ in $g$, a delay occurs in receiving information from each other. The delay cost can be represented by the distance between them in $g$ which is defined by the number of links in

---

[1]     In this paper, we use the terms "graph" and "network" interchangeably.
[2]     By deletion of a node, we mean that all the incident links to the node are deleted together.

the shortest path between $i$ and $j$ in $g$ and denoted by $d(i,j;g)$. If there is no path between $i$ and $j$ for some $i,j \in N$, we define $d(i,j;g) = \infty$.

The total cost that each agent $i$ incurs in graph $g$ is the sum of the delay cost and the connecting cost, which is given by

$$w_i(g) = T_i(g) + C_i(g),$$

where $T_i(g) \equiv \sum_{j \neq i} d(i,j;g)$ and $C_i(g) \equiv \sum_{j:ij \in g} c$. We will call $T_i(g)$ and $C_i(g)$ agent $i$'s delay status and connecting status respectively. Note that $w_i(g) = \infty$ for any $i \in N$ if $g \notin \mathbb{C}$, while $w_i(g) < \infty$ for any $i \in N$ if $g \in \mathbb{C}$. Note that if an agent adds a link, it has a tradeoff between increasing his total linking cost and reducing the total delay cost which is the benefit of adding a link.

We consider the following model of dynamic network evolution. In each period $t$, player $t$ is born. Thus, in the first period, the population consists of only one agent. In the $t$-th period, player $t$ joins the population consisting of $t-1$ players. The stage game of the $t$-th period proceeds as follows: First, player 1 decides whether to propose connecting a link with the newcomer, player $t$.[3] After the decision of player $t$ whether to accept the proposal, player 2 decides whether to propose a link with player $t$ and so on until player $t-1$ makes a decision. Players are forward-looking only within a period. In other words, players make backward inductions in a stage game of each period. When they make a decision at period $t$, they do not take into account the possibility of repeating this procedure against player $t+1, t+2, \cdots$. Once a link is formed, the connecting cost $c$ is sunk, meaning that no agent considers the option of severing an already formed link because a player could not recover $c$ by doing so. On the other hand, the communication costs are not sunk. Communications occur in every period. Thus, the delay costs are incurred in each period. For example, if $g_{t+1} = g_t + t(t+1)$, player $t$ incurs delay costs $T_t(g_t)$ at both period $t$ and period $t+1$ and further incurs the additional delay cost of one for communication with a new entrant $t+1$ at period $t+1$.

## 4. Analysis

In this section, we analyze the dynamic model with the constraint that the network is required to be connected.[4] We consider both the case of no fault and the case of faults.

### 4.1. No Fault Case

First, we consider the case that there is no possibility that any node will be destroyed. Note that the network at each period $t$ must be subgame perfect due to the prescribed sequential nature of a stage game. By a subgame perfect network, we mean a network that is realized in a subgame perfect equilibrium in the $t$-th stage game. Thus, under the requirement that a network be connected, i.e., $g \in \mathbb{C}$, a dynamically $\mathscr{C}$-stable network at period $t$ is defined by $g_t \in \mathbb{C}$ such that (i) $g_t$ is a subgame perfect network and (ii) $w_k(g_t) \leq w_k(g_{t-1})$, $\forall k \in S(\Delta g)$ where $\Delta g = g_t - g_{t-1}$. If $g_t$ is a dynamically $\mathscr{C}$-stable network at period $t$ for all $t$, we call $\{g_t\}_{t=1}^{\infty}$ dynamically $\mathscr{C}$-stable. We will use dynamic $\mathscr{C}$-stability as our main solution concept. Additionally, note that a sequence of $\mathscr{C}$-stable networks at period $t$ is strictly increasing in the sense that $g_t \subset g_{t'}$ for any $t, t' \ni t' > t$, because the assumption that the cost of linking in a period becomes sunk in the next period thereafter eliminates the possibility that an existing link is severed. This concept of $\mathscr{C}$-stability is based on pairwise stability by Jackson and Wolinsky [2] that requires a new link to be formed by mutual consent of the linking agents.[5]

It is worthwhile to mention the relations of our concept to other concepts in literature. Our definition of dynamic $\mathscr{C}$-stability is distinguished from that of Watts [9] and Jackson and Watts [10] who assume that players are myopic. In our definition, players are farsighted in the sense that they make linking decisions by comparing $g_{t-1}$ with $g_t$ that will be completed at the end of period $t$.

---

3   Since he is the most senior in the society, he is entitled to move first.
4   Requiring connectivity is reasonable, especially when pieces of information possessed by each agent are complementary.
5   Severing an existing link unilaterally does not occur in $\mathscr{C}$-stable networks due to the assumption of sunk linking costs.

In other words, player $i$ does not compare $g_{t-1}$ with $g_{t-1} + it$ because he is farsighted enough to expect other links to follow in period $t$. However, they are not perfectly farsighted in the sense that they do not compare $g_{t-1}$ with $g_{t+1}$, $g_{t+2}$ and $\cdots$. Thus, we call our definition of dynamic stability intraperiod-farsighted stability to distinguish it from farsighted stability by Chwe [11] and Page et al. [12].

To characterize dynamic $\mathscr{C}$-stable networks, we need several lemmas. Lemma 1 tells us whether two extreme players (the most senior player and the most junior player) in a line network have an incentive to link between them.

**Lemma 1.** *For any given $t$, define $I_t \equiv (a_t, b_t]$ by*

$$
I_t = \begin{cases} \left( \frac{(t-1)(t-3)}{4}, \frac{(t-1)^2}{4} \right] & \text{for odd } t \geq 3 \\[2mm] \left( \frac{(t-2)^2}{4}, \frac{t(t-2)}{4} \right] & \text{for even } t \geq 4 \end{cases}
$$

*Then, (i) $a_{t+1} = b_t$, (ii) $I_t \cap I_{t'} = \varnothing$ for $t \neq t'$, and (iii) $\bigcup_{t=3}^{\infty} I_t = \mathbb{R}^+$.*

This lemma says that $\{I_t\}$ is a partition of $\mathbb{R}^+$ that consists of a monotonically increasing sequence of intervals. The interval $I_t$ is essential to characterizing the stable network, because $a_t$ and $b_t$ have the interpretations as the benefits from node 1 linking with node $t-1$ and node $t$, respectively. Intuitively, if $c \in I_t$, node 1 will have no incentive to link with node $t-1$ at period $t-1$ because $c > a_t$ but he will have an incentive to link with node $t$ at period $t$ because $c < b_t$, assuming that $g_{t-1}$ is a line.

To understand the intuition, let $\Delta T_k$ be the benefit of node $k$ from linking with the new node $t$ at period $t$. Then, the benefit is the difference in his delay status. Assuming that $g_{t-1}$ is a line, the incentive of node 1 to link with node $t$ is exactly identical to the incentive of node $t-1$ to link with node $t$. Hence, $1t \in g_t$ implies $(t-1)t \in g_t$.[6] Thus, $b_t = \Delta T_1 = T_1(\bar{g}_t) - T_1(\bar{g}_t + 1t)$ where $\bar{g}_t = g_{t-1} + (t-1)t$. If $t = 2l + 1$ is odd,

$$
\begin{aligned}
\Delta T_1 &= \sum_{j=1}^{t-1} j - \sum_{j=1}^{l} 2j \\
&= \frac{t(t-1)}{2} - l(l+1) \\
&= \frac{t(t-1)}{2} - \frac{t-1}{2} \frac{t+1}{2} \qquad (\because l = \frac{t-1}{2}) \\
&= \frac{(t-1)^2}{4},
\end{aligned} \tag{1}
$$

and if $t = 2l$ is even,

$$
\begin{aligned}
\Delta T_1 &= \sum_{j=1}^{t-1} j - \left( \sum_{j=1}^{l-1} 2j + l \right) \\
&= \frac{t(t-1)}{2} - [l(l-1) + l] \\
&= \frac{t(t-1)}{2} - \frac{t^2}{4} \qquad (\because l = \frac{t}{2}) \\
&= \frac{t(t-2)}{4}.
\end{aligned} \tag{2}
$$

---

6    For a formal proof, see Lemma 3.

Similarly, $a_t$ can be computed by replacing $t$ by $t-1$ in (1) and (2).

It is intuitively clear that node 1 has more incentive to link with the entrant as the length of the line network is longer. Thus, Lemma 2 follows.

**Lemma 2.** *Assume that $g_{t-1}$ is a line; i.e., $g_{t-1} = \{12, 23, \cdots, (t-2)(t-1)\}$. Then, $\Delta T_k$ is strictly decreasing in k, for $k = 1, \cdots, t-2$.*

This lemma implies that if node 1 has no incentive to link with node $t$, neither does node $k$ for $k = 2, \cdots, t-2$. The following lemma is helpful for understanding the incentive of node $t-1$ to link with node $t$.

**Lemma 3.** *Assume that $g_{t-1}$ is a line. Then, $1t \in g_t$ implies $(t-1)t \in g_t$.*

This lemma implies that the incentive of node $t-1$ to link with the entrant $t$ in a line network $g_{t-1}$ is congruent with the incentive of node 1.

**Lemma 4.** *For given t, if $c \notin \bigcap_{k=3}^{t-1} I_k$, i.e., $c > a_t$, $g_{t-1}$ is a line.*

The main insight behind this lemma is that the most senior player enjoys the first mover advantage. He can save the connecting cost by taking advantage of juniors who must link with the newcomers. We can call this the deferral principle.[7] The first mover (senior) advantage is mainly due to the requirement that the network must be connected.

To illustrate $\mathscr{C}$-stable networks, consider $t = 4$. It is clear that $g_2 = \{12\}$, since $g_2 \in \mathbb{C}$. At $t = 3$, node 1 would not link with node 3 if $c > 1$, because he knows that if he does not, node 2 must link to node 3. Accordingly, $g_3 = \{12, 23\}$ for any $c > 1$. This is contrasted with the case that $c < 1$ in which node 1 will link to the new node 3 even if he knows that a direct link between node 2 and node 3 will follow his linking with node 3 because the benefit of reducing a delay by the direct link with node 3 exceeds the linking cost ($c < 1$). At $t = 4$, node 3 must link with the new node 4 (i.e., $34 \in g_4$) if $4i \notin g_4$ for all $i = 1, 2$. If $24 \in g_4$, node 3 will not link with node 4 (i.e., $34 \notin g_4$), as far as $c > 1$. If $14 \in g_4$, $34 \in g_4$ if $1 < c < 2$ because the linking cost $c$ is smaller than the benefit of reducing delay which is 2. If $c > 2$, $14 \in g_4$ would lead to $34 \notin g_4$ and $24 \notin g_4$. Therefore, node 1 would prefer not to link to node 4, because in that case, node 3 must link with node 4 and this is better for node 1. Thus, $14 \in g_4$ if and only if $1 < c < 2$. The resulting stable network at $t = 4$ is a circle if $1 < c < 2$ and a line if $c > 2$. This argument can be generalized for $t > 4$ by the following proposition.

**Proposition 1.** *If $c < 1$, the complete graph is the only dynamically $\mathscr{C}$-stable network. For any $c > 1$, there exists a finite time $\bar{t}(c)$ such that, for all $t < \bar{t}(c)$, $g_t$ is a line, and $g_t$ is a circle if $t = \bar{t}(c)$. In particular, $\bar{t}(c)$ is determined by t such that $c \in I_{\bar{t}(c)}$, and $\bar{t}(c)$ is strictly increasing in c.*

In Appendix B, we will provide an algorithm which we call optimal deferral algorithm to find the $\mathscr{C}$-stable network for any $t$.

By varying $c$ instead of varying $t$ and fixing $c$, Proposition 1 can be interpreted as follows.

---

[7] The insight of the deferral principle is similar to the principle of last clear chance in law. In tort law, if several legal parties are jointly responsible for an accident but a second mover could have avoided the accident, the second mover tends to be placed in a disadvantageous position by the contributory negligence rule. It is referred to as the doctrine of last clear chance. In our model, all the players are jointly responsible for network disconnection. In this case, the last mover bears the burden of connecting with the entrant.

**Corollary 1.** *Suppose that $c \in I_{\bar{t}}$ for some $\bar{t}$. Then, $g_t$ is a circle if $t = \bar{t}$ ($c \in I_t$), and $g_t$ is a line if $c \notin \bigcup_{k=3}^{t} I_k$.*

We can compare the result in this dynamic model of network formation with that of a static model that was obtained in Jun and Kim [1]. First, the complete network is uniquely $\mathscr{C}$-stable if $c < 1$ in both the dynamic model and the static model. Second, the line network is statically $\mathscr{C}$-stable if $c > \frac{k(k-2)}{4}$ for even $k$ and $\frac{(k-1)^2}{4}$ for odd $k$. Proposition 1 shows that the line network is dynamically $\mathscr{C}$-stable in the same range of $c$. Third, the circle network is dynamically $\mathscr{C}$-stable if $c \in I_k$. Let $\bar{I}_k$ be the range of $c$ in which a circle is $\mathscr{C}$-stable in a static model. Then, we have

$$
\bar{I}_k = \begin{cases} \left( \frac{(k-1)(k-3)}{8}, \frac{(k-1)^2}{4} \right] & \text{if } k \text{ is odd} \\ \left( \frac{k^2 - 4k + 8}{8}, \frac{k(k-2)}{4} \right] & \text{if } k = 4l \\ \left( \frac{(k-2)^2}{8}, \frac{k(k-2)}{4} \right] & \text{if } k = 4l - 2, \end{cases} \tag{3}
$$

by Proposition 1 of Jun and Kim [1]. Since $I_k \subset \bar{I}_k$ for all $k (\geq 4)$, the dynamic $\mathscr{C}$-stability of a circle implies static $\mathscr{C}$-stability, but not vice versa.

*4.2. Fault Case*

Now, we consider the possibility that $q (\geq 1)$ nodes are destroyed due to exogenous shocks, for example, an enemy's attack.[8] Thus, in this section, besides connectivity, we impose a further requirement that a graph remains connected even after the deletion of any at most $q$ nodes and their direct links. Then a graph $g_t \in \mathbb{C}^q$ at period $t$ is dynamically $\mathscr{C}^q$-stable if and only if (i) $g_t$ is subgame perfect and (ii) $w_k(g_t) \leq w_k(g_{t-1})$, $\forall k \in S(\Delta g)$.

**Proposition 2.** *If $c < 1$, the complete graph is the only $\mathscr{C}^q$-stable network. For any $q \geq 1$, if $t \leq q + 2$, the unique dynamically $\mathscr{C}^q$-stable network is the complete graph. If $t > q + 2$, it must be a graph such that there are $q + 2$ centers called hubs forming a complete subnetwork among them and the rest of the nodes have $(q + 1)$ links to the centers.*

Without the possibility of node extinction, hubs would not be necessary for stability, but hubs are essential for stability in the presence of a possibility of attacks on nodes, and furthermore, more hubs are necessary in the presence of attacks on more nodes (larger $q$).

It is easy to see that a dynamically $\mathscr{C}^q$-stable network is, in general, not $\mathscr{C}^q$-stable in a static model. Dynamic $\mathscr{C}^q$-stability requires $q + 2$ centers that form a clique.[9] If a pair of centers delete the link between them, it is profitable if $c > 1$, because the delay cost per each node increases by one and $c > 1$. Therefore, if $c > 1$, no dynamically $\mathscr{C}^q$-stable network is $\mathscr{C}^q$-stable in a static model. This disparity mainly comes from our assumption that $c$ is sunk. Once a link is formed in a previous period, it is never severed in our dynamic model, even if it may become redundant by subsequently formed links. Thus, the resulting dynamically $\mathscr{C}^q$-stable network may not be $\mathscr{C}^q$-stable in a static sense because such a link may not be formed in a static model. To elaborate, note that all nodes must have at least $q + 1$ degrees in a $q$-connected network (see Lemma 2 of Jun and Kim [1]). In our dynamic model, $q + 2$ centers form $q + 1$ links with all the other center nodes and then the $(q + 3)$-th node form links with $q + 1$ center nodes. Let $i_1$ be the remaining center node; i.e., $i_1(q + 3) \notin g_{q+3}$. Similarly, if $(q + 4)$-th node forms links with center nodes except node $i_2(\neq i_1)$, it is dynamically $\mathscr{C}^q$-stable. It is, however, not statically

---

[8]    In the static model of Jun and Kim [1] in which the number of players is fixed, it is natural to adopt the interpretation that nodes naturally die with some exogenous probability, but in this dynamic model, it is less natural because an exogenously fixed probability of death must imply that $q$ increases as the population size grows. Thus, in this paper, we adopt the interpretation of node destruction as a result of an enemy's attack. An exogenous $q$ in this dynamic model can be justified by the implicit assumption that the enemy's ability to attack is limited only to $q$ nodes.

[9]    A clique is defined by a maximal complete subgraph of a given graph.

$\mathcal{C}^q$-stable, because two other center nodes $i_3, i_4$ for $i_3, i_4 \neq i_1, i_2$ could benefit from severing their direct link if $c > 1$, while the remaining graph is still $q$-connected, although severing the link is not permissible in the dynamic model in which the linking cost is sunk.

Clearly, there is a coordination problem,[10] because no player would want to be a hub if hubs are more likely to be attacked.[11] In our dynamic model, however, such a coordination problem is easily resolved by seniority. The most senior players must form the hubs to maintain $q$-connectivity. In this sense, the possibility of external attacks gives seniors a disadvantage to becoming a hub.

## 5. Conclusions

In this paper, we characterized the $\mathcal{C}$-stable network and the $\mathcal{C}^q$-stable network in a dynamic model in which players enter the population sequentially. We believe that it is worthwhile to examine how robust our characterization is to variations of the model. We look forward to more enriched models in this direction.

**Author Contributions:** Formal analysis, T.J. and J.-Y.K.; writing–original draft preparation, J.-Y.K.; writing–review and editing, J.-Y.K. All authors have read and agreed to the published version of the manuscript.

**Funding:** This research received no external funding.

**Conflicts of Interest:** The authors declare no conflict of interest.

## Appendix A. Proofs

**Proof of Lemma 1.** (i) If $t$ is odd, $a_{t+1} = \frac{(t-1)^2}{4} = b_t$. If $t$ is even, $a_{t+1} = \frac{(t+1-1)(t+1-3)}{4} = \frac{t(t-2)}{4} = b_t$. (ii) and (iii) are clear from (i). $\square$

**Proof of Lemma 2.** We have

$$
\begin{aligned}
\Delta T_k &= T_k(\bar{g}_t) - T_k(\bar{g}_t + kt) \\
&= T_1(\bar{g}_{t-k+1}) - T_1(\bar{g}_{t-k+1} + 1(t-k+1)),
\end{aligned}
\tag{A1}
$$

by relabeling $k, \cdots, t$ to $1, \cdots, t-k+1$, where $\bar{g}_t = g_{t-1} + (t-1)t = \{12, 23, \cdots, (t-1)t\}$, for any $k = 2, 3, \cdots, t-1$. Since $\bar{g}_t + 1t$ is a circle with $t$ nodes for any $t$, this is strictly increasing in $t-k+1$ (circle size), so it is strictly decreasing in $k$. $\square$

**Proof of Lemma 3.** Consider the incentive of node 1 to link with node $t$. If he does not link, neither does node $k(\leq t-2)$ by Lemma 2. Thus, node $t-1$ must link with node $t$ since $g_t \in \mathcal{C}$. In this case, $g_t$ is a line such that $g_t = \{12, 23, \cdots, (t-1)t\}$, and node 1 becomes the extreme node of the line network. On the other hand, if node 1 links with node $t$, node $t-1$ will link with node $t$, too, because node $t-1$ would otherwise become an extreme node of a line network $g_t = \{t1, 12, \cdots, (t-2)(t-1)\}$. Therefore, if node 1 prefers a circle to a line (with himself being an extreme node of the line), node $t-1$ also prefers a circle to a line in which he is one of the extreme nodes. This implies that node $t-1$ prefers to link with node $t$, whenever node 1 prefers to link with node $t$. $\square$

**Proof of Lemma 4.** Since $c > a_t$, we have $c > a_k$ for any $k = 3, \cdots, t-1$ by Lemma 1. This implies that node 1 has no incentive to link with the entrant $k$ for $k = 3, \cdots, t-1$. Then, by Lemma 2, node $j$ has no incentive to link with the entrant $t-1$ for any $j = 3, \cdots, t-3$, either. Thus, connectivity requires node $t-2$ to link with node $t-1$. (I.e., node $t-2$ has an incentive to link with node $t-1$, considering $w_i(g) = \infty$ if $g$ is not connected). This implies that the resulting network $g_{t-1}$ is a line. $\square$

---

**Proof of Proposition 1.** It is trivial that the complete graph is the only dynamically $\mathscr{C}$-stable network if $c < 1$. For $c > 1$, suppose $c \in I_{\bar{t}(c)} \equiv (a_{\bar{t}}, b_{\bar{t}})$. Then, it is clear from Lemma 1 that $a_t < c$, $\forall t < \bar{t}(c)$.

(i) We will show that for any $t < \bar{t}(c)$, no node $k(< t - 1)$ has an incentive to form a link with a new entrant $t$. By Lemma 4, $g_{t-1}$ is a line. Now, since $c > a_t$, node 1 has no incentive to link with node $t$. Then, by Lemma 2, node $k$ for $k = 2, \cdots, t - 2$ will not link with node $t$, implying that node $t - 1$ must link with node $t$ due to the requirement of connectivity. Therefore, $g_t$ is a line for any $t < \bar{t}(c)$.

(ii) We will show that $g_{\bar{t}(c)}$ is a circle. Since $c \in I_{\bar{t}(c)}$, $c < b_t$. This implies that node 1 has an incentive to link with node $t$. Note that node 2 has no incentive to link with node $t$, because $T_2(\bar{g}_t) - T_2(\bar{g}_t + 2t) = T_1(\bar{g}_{t-1}) - T_1(\bar{g}_{t-1} + 1(t - 1)) = a_{t-1} < c$. Moreover, by Lemma 4, no node $k = 3, \cdots, t - 2$ has an incentive to link with node $t$. The incentive of node $t - 1$ to link with node $t$ is identical with node 1's incentive. Additionally, the incentive of node $t$ to connect two links instead of one link is exactly identical with the incentives of node 1 and node $t - 1$. Thus, $(t - 1)t \in g_t$. Therefore, $g_t$ is a circle.

(iii) Since both $b_t$ increases in $t$ and $\lim_{t \to \infty} b_t = \infty$, for any $c < \infty$, there exists $\bar{t}(c) < \infty$ such that $c \in I_{\bar{t}(c)}$; i.e., $g_{\bar{t}}$ is a circle and $g_t$ is a line for any $t < \bar{t}$. $\square$

**Proof of Proposition 2.** (i) It is trivial that the complete graph is the only $\mathscr{C}^q$-stable network if $c < 1$.

(ii) It is clear that $g_t$ is a complete graph for any $t \leq q$. (If $ij \notin g_t$, deleting $(t - 2)$ nodes except $i$ and $j$ would make it disconnected. Thus, $g_t \notin \mathbb{C}^q$.)

(iii) We can show that $g_t$ is a complete graph for $t = q + 1, q + 2$. If $t_0 t \notin g_t$ for $t = q + 1$ or $q + 2$ and for some $t_0 \leq q$, deleting all other nodes than node $t_0$ and node $t$ would make the graph disconnected.

(iv) If $t > q + 2$, we will first show that $g_t \in \mathbb{C}^q$. Let $T = \{1, 2, \cdots, t\}$, $H = \{1, 2, \cdots, q + 2\}$ and $J = \{q + 3, \cdots, t\}$. We may call $i \in H$ an insider and $j \in J$ an outsider. Let $S$ be any subset of $T$ with $|S| = q$. We will denote the graph obtained by deleting all the nodes in $S$ from $g_t$ by $g_t(T \setminus S)$. Now, if $S \subset H$, two nodes will remain in hub $H$. Let the nodes be $i$ and $i'$. Then, $ii' \in g_t(T \setminus S)$ because $ii' \in g_t(T)$ and $i, i' \notin S$. Since any outsider $j$ has $(q + 1)$ links with insiders, $ij \in g_t(T)$ or $i'j \in g_t$. If $ij \notin g_t(T)$ and $i'j' \notin g_t(T)$ for any $j, j' \in J$, it implies that $i'j, ij' \in g_t(T)$, so there is a path between $j$ and $j'$, $j \to i' \to i \to j'$ in $g_t(T - S)$. If $ij, ij' \notin g_t(T)$ for the same $i$, there is a path between $j$ and $j'$, $j \to i' \to j'$ in $g_t(T - S)$. If $j \in S$ or $j, j' \in S$, it is trivial that $g_t \in \mathbb{C}^q$.

(v) It remains to show that $g_t$ is $\mathscr{C}^q$-stable. First, any outsider $j \in J$ would not connect more than $q + 1$ nodes with insiders from the assumption that $c > 1$, since additional link will reduce delay cost by one. Second, any outsider could not have less than $q + 1$ nodes with insiders. If $ij, i'j \notin g_t(T)$ for some $j \in J$, choose $S = H \setminus \{i, i'\}$. Then, there is no path between $i$ and $j$ in $g_t(T \setminus S)$, i.e., $g_t \notin \mathbb{C}^q$. Therefore, in a $\mathscr{C}^q$-stable network, any outsider $j \in J$ must have exactly $q + 1$ links with nodes in $H$. $\square$

## Appendix B. Optimal Deferral Algorithm

**Lemma A1.** *For given $t$, if $c \in I_t$, a necessary condition for $g_t$ to be $\mathscr{C}$-stable is that $d(i, j; g_t) < t - 1$ for all $i, j \leq t$.*

**Proof.** Suppose that $d(i, j; g_t) \geq k(\geq t - 1)$ for some $i, j \leq t$. Let the shortest path between them be $\{i_0, i_1, i_2, \cdots, i_k\}$ with $i_0 \equiv i$ and $i_k \equiv j$. We can relabel $i_0, \cdots, i_k$ by $1, 2, \cdot, k + 1$. Since $c \in I_t = (a_t, b_t]$ and $t \leq k + 1$, we have $c < a_{k+1}$ by Lemma 1. This implies that one extreme node 1 (or node $i$) has an incentive to link with the other extreme node $k + 1$ (or node $j$) directly or indirectly via a more junior node. Thus, it must be that $d(i, j; g_t) < k$. This is a contradiction. $\square$

Optimal Deferral Algorithm:

If $c \in I_t$ for some $t$, $g_t$ is a circle, while $g_k$ is a line for any $k \leq t - 1$ by Proposition 1. Now, the following steps will generate $g_l$ for any $l > t$.

Step 1:

In period $l$, link players $l$ and $l-1$ and let $g_l^0 = g_{l-1} + l(l-1)$. Define $S_1^l = \{i < l-1 \mid d(i,l;g_l^0) \geq t-1\}$. If $S_1^l = \varnothing$, define $g_l = g_l^0$. If $S_1^l \neq \varnothing$, let $i_1^* = \sup S_1^l$ and define $g_l = g_l^0 + i_1^* l$.

Step 2:

For $j \geq 1$, let $S_{j+1}^l = \{i \in S_j^l \setminus \{i_1^*\} \mid d(i,l;g_l^j) \geq t-1\}$. If $S_{j+1}^l = \varnothing$, $g_l = g_l^j$. If $S_{j+1}^l \neq \varnothing$, let $i_{j+1}^* = \sup S_{j+1}^l$ and define $g_l^{j+1} = g_l^j + i_{j+1}^* l$.

Step 3:

Repeat step 1 and step 2 until $S_{j*}^l = \varnothing$. Define $g_l = g_l^{j*}$.

We will sketch the proof of why the network that is obtained by the optimal deferring algorithm is dynamic $\mathscr{C}$-stable.

Start from period $t$ such that $g_t$ is a circle, i.e., $c \in I_t$. In step 1, for $l = t+1$, $S_1^{t+1}$ is the set of nodes $i$ such that the (shortest) distance from $t+1$ in $g_{t+1}^0$ is not less than $t-1$. If $S_1^{t+1} = \varnothing$, i.e., $d(i,t+1;g_{t+1}^0) < t-1$ for all $i \leq t$, $g_{t+1}$ need not have other link than $t(t+1)$, because $c \in I_t$. So, $g_{t+1} = g_t + t(t+1)$ is $\mathscr{C}$-stable. If $S_1^{t+1} \neq \varnothing$, the nodes $i \in S_1^{t+1}$ needs an additional link to maintain $d(i,t+1) < t-1$ by Lemma 4. By the senior advantage, the most junior, i.e., $i_1^* = \sup S_1^{t+1}$ takes the role of linking with $t+1$. We denote the resulting network by $g_{t+1}^1 = g_{t+1}^0 + i_1^*(t+1)$.

Now, it is still possible that $d(i,t+1;g_{t+1}^1) \geq t-1$ for some $i$. It is the reason why we need step 2. In step 2, define $S_2^{t+1} = S_1^{t+1} \setminus \{i_1^*\}$ by the set of nodes $i$ in $S_1^{t+1} \setminus \{i_1^*\}$ such that the shortest distance from $t+1$ in $g_{t+1}^1$ is not less than $t-1$. If $S_2^{t+1} \neq \varnothing$, take $i_2^* = \sup S_2^{t+1}$ again by the senior advantage and link $i_2^*$ to $t+1$. We define the resulting network by $g_{t+1}^2 = g_{t+1}^1 + i_2^*(t+1)$.

We repeat this process for any $j > 2$ and for any $l > t+1$, until $S_j^l = \varnothing$. If we define $j^*$ by $j$ such that $S_j^l = \varnothing$, it is clear that $g_l = g_l^{j*}$ is $\mathscr{C}$-stable.

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
