# Peer review of "A Note on Connectivity and Stability in Dynamic Network Formation"

_games, doi:10.3390/g11040049_

Round 1

Reviewer 1 Report

The paper studies the following model. Each vertex in a graph corresponds to a player. Players arrive sequentially. At time step t, Player t arrives and all the existing players need to decide, sequentially starting from 1 up to t-1, whether they form an edge with Player t. There is a fixed one-time cost c for forming an edge. Also, player i pays the sum of distances to every other player j in the formed graph, where the cost of a non-connected graph is infinity. The choices of the players at turn t are found by reasoning only about the future choices of players at step t, which can be computed using a standard backwards induction. I find this model nice.   The main result in the paper shows that the real numbers can be partitioned into intervals I_1, I_2,… Suppose the cost c of forming an edge is in I_t, then all graphs before step t are lines and at step t a circle is formed. The result is nice, though I think the treatment of the model is a bit superficial and can be studied further: e.g., either than the two ranges of c that are studied (c<1 and c in I_t), what graphs are formed?    The paper then continues to study the same setting only when players can disappear from the network.    The proofs seem correct but I think the writing can be improved. For example, the claim "1t in g_t implies (t-1)t in g_t” at the top of page 5 can be made a lemma and proven formally. Also g_\infy in the middle of page 4 is a weird notation; I would prefer either … or g_n.  Page 3, second paragraph: the d(i,j;g) notation appears for the first time. Shouldn’t it be d(i,j)? Prop 1: “i.e., c \in I_t” is redundant It might help to move the discussion after corollary 1 earlier. It helps in understanding the model.   

A similar model is called “network creation game” and I think the paper would benefit from a comparison to the line of work initiated by:   Alex Fabrikant, Ankur Luthra, Elitza N. Maneva, Christos H. Papadimitriou, Scott Shenker: On a network creation game. PODC 2003   Also, somewhat related is the following paper, which takes a dynamic approach to classic one-shot games:   Renato Paes Leme, Vasilis Syrgkanis, Éva Tardos: The curse of simultaneity. ITCS 2012

Reviewer 2 Report

This paper is a theoretical analysis of a network formation problem, where the network may be attacked. The objective is that the network remains connected even after the attack. A novel concept of stability is introduced to capture this objective.

Comments to the author:

Major Comments:

  • Introduction: I think the introduction needs some work. While indeed you are defining a new stability concept, plenty of work has been done with respect to network formation under the threat of disruption/attack that you neglect to mention here and that are definitely related to your research. Below a (non-exhaustive) list. To me a better discussion of the literature and your addition to it is needed.
    • Goyal, S., & Vigier, A. (2014). Attack, defence, and contagion in networks. The Review of Economic Studies81(4), 1518-1542.
    • Haller, H., & Hoyer, B. (2019). The common enemy effect under strategic network formation and disruption. Journal of Economic Behavior & Organization162, 146-163.
    • Hoyer B, De Jaegher K. Strategic Network Disruption and Defense. Journal of Public Economic Theory. 2016;18(5):802-830. doi:10.1111/jpet.12168
    • DziubiÅ„ski, M., & Goyal, S. (2013). Network design and defence. Games and Economic Behavior79, 30-43.
  • Model: I think some of your modeling decisions need to be made clearer so that one can follow the model. E.g.
    • Can connecting cost and delay cost be any value larger 0? Are there no boundaries?
    • Do both agent i and j incur the cost for the link?
    • I can see how the connecting costs are considered as sunk costs and are only incurred once. However, what about the delay costs? If I am linked to node t and in period t+1 node t links to node t+1, I suddenly incur delay costs, no? And in the following period through that link I may incur further delay costs if node t+1 links to node t+2? So are the delay costs per period or also sunk? If they are not per period, what happens if delay costs are decreased? Consider the following: I am linked to node t, who is linked to node t+1, who is linked to node t+2, who is linked to node t+3. In period t+4 node t+4 joins and (for whatever reason) node both node t+3 and me link to node t+4. Then my delay cost to node t+3 decreases as my shortest distance to node t+3 is no 2 instead of 3.
    • What are the benefits of linking? In your model, you explicitly model costs but not the benefits. So why should anyone link to anyone else?
  • Analysis: I am unfortunately missing some intuition here as well.
    • Lemma 1 comes out of nowhere. Some intuition for what this interval actually signifies and how it plays a role in the stability would be nice.
    • What about the incentives of the entrant to accept a link?
    • You are basing your argumentation below Corollary 1 on costs larger or smaller than 1. So does that mean the benefit I get from being linked to another player is 1? If so, please specify in the modelling section.
  • Fault Case:
    • How do nodes die? Are they specifically attacked? Are they randomly dying? In the case of a specific attack, the hubs would be attacked. This would have to be taken into account in the decisions of the nodes. So no node would want to be a hub.
    • Please elaborate on how nodes dies, otherwise I don’t think one can fully follow your results.

Minor Comments:

  • Typo: Section 3, second line: of connecting each of his direct links (not link)
  • Typo: Section 3, page 3: The stage game of the t-th period proceeds as follows
  • Typo: Section 4, page 4: let $\Delta$ T_k be the benefit of node k

Round 2

Reviewer 2 Report

I think the paper has been much improved as compared to the previous version. It now reads as a complete paper and no longer like a note. Thanks for answering my questions and clearing up the issues I had previously.